# Structure-guided design of pure orthosteric inhibitors of αIIbβ3 that prevent thrombosis but preserve hemostasis

Brian D. Adair[1,2,3], José L. Alonso[1,2,3], Johannes van Agthoven[1,2,3], Vincent Hayes[4], Hyun Sook Ahn[4], I-Shing Yu [5], Shu-Wha Lin[6], Jian-Ping Xiong[1,2,3], Mortimer Poncz[4] & M. Amin Arnaout [1,2,3]*

A prevailing dogma is that inhibition of vascular thrombosis by antagonizing platelet integrin αIIbβ3 cannot be achieved without compromising hemostasis, thus causing serious bleeding and increased morbidity and mortality. It is speculated that these adverse outcomes result from drug-induced activating conformational changes in αIIbβ3 but direct proof is lacking. Here, we report the structure-guided design of peptide Hr10 and a modified form of the partial agonist drug tirofiban that act as "pure" antagonists of αIIbβ3, i.e., they no longer induce the conformational changes in αIIbβ3. Both agents inhibit human platelet aggregation but preserve clot retraction. Hr10 and modified tirofiban are as effective as partial agonist drugs in inhibiting vascular thrombosis in humanized mice, but neither causes serious bleeding, establishing a causal link between partial agonism and impaired hemostasis. Pure orthosteric inhibitors of αIIbβ3 may thus provide safer alternatives for human therapy, and valuable tools to probe structure–activity relationships in integrins.

[1] Leukocyte Biology & Inflammation Program, and Structural Biology Program, Massachusetts General Hospital, Boston, MA 02114, USA. [2] Division of Nephrology, Department of Medicine, Massachusetts General Hospital, Boston, MA 02114, USA. [3] Harvard Medical School, Boston, MA 02115, USA. [4] Division of Hematology, The Childrens Hospital of Philadelphia, Philadelphia, PA 19104, USA. [5] Laboratory Animal Center, College of Medicine, National Taiwan University, Taipei, Taiwan. [6] Department of Clinical Laboratory Sciences and Medical Biotechnology, National Taiwan University, Taipei, Taiwan. *email: aarnaout1@mgh.harvard.edu

Platelet activation and accumulation at the site of blood vessel injury are the initial steps in hemostasis. When activated by several agonists, including adenosine diphosphate (ADP), thrombin, or collagen, platelets adhere to the disrupted surface, and aggregate upon binding of soluble fibrinogen (FB) or other proteins[1] to agonist-activated αIIbβ3[2]. Fibrin generated by thrombin at or near the platelet surface also binds αIIbβ3, driving clot retraction[3], thereby consolidating the integrity of the hemostatic plug, restoring blood flow, and promoting wound closure[4]. Excessive platelet activation by agonists may lead to the formation of occlusive thrombi, which are responsible for acute myocardial infarction and stroke[5], hemodialysis access failure[6], early loss of kidney allograft[7] and tumor growth, and metastasis[8].

The three parenteral anti-αIIbβ3 drugs eptifibatide, tirofiban, and abciximab have demonstrated efficacy in reducing death and ischemic complications in victims of heart attacks[9]. However, their clinical use in acute coronary syndrome has been associated with serious bleeding, which often requires cessation of therapy, putting heart attack victims at high risk of rethrombosis. And oral anti-αIIbβ3 agents given to patients at risk of acute coronary syndrome were abandoned because of the increased risk of patient death linked to paradoxical coronary thrombosis[10,11]. Concluding that the adverse outcomes resulting from targeting αIIbβ3 are unavoidable, pharmaceutical companies developed inhibitors of the platelet ADP receptor $P_2Y_{12}$ and the thrombin receptor PAR1, both acting upstream of αIIbβ3. However, a considerable number of patients receiving these newer drugs continue to experience serious bleeding and thrombotic events[12]. Thus, there remains an unmet clinical need for new antithrombosis drugs that maintain efficacy while preserving hemostasis[13].

Previous studies have shown that the three current anti-αIIbβ3 drugs are partial agonists, i.e., they trigger large activating conformational changes in αIIbβ3 that enhance receptor binding to physiologic ligand, promoting thrombosis[14]. These drugs also induce neoepitopes for natural antibodies, causing immune thrombocytopenia[15]. X-ray structures of unliganded and ligand-bound integrins[16–18] showed that the conformational changes in

the integrin induced by binding of ligands or ligand-mimetic drugs are initiated in the integrin ligand-binding vWFA domain (A- or I domain). Ligand binding to β3 integrins triggers tertiary changes in the A domain of the β3 subunit (βA domain), comprising the inward movement of the N-terminal α1 helix (reported by movement of βA-$Y^{122}$) toward the $Mg^{2+}$ or $Mn^{2+}$ ion coordinated at the metal ion-dependent adhesion site (MIDAS). This reshapes the C-terminal F-α7 loop and repositions the α7 helix causing a swing-out of the hybrid domain underneath the βA domain, which converts the integrin from the genu-bent to the genu-extended conformation, separates the transmembrane and cytoplasmic tails, allowing formation of the integrin–cytoskeleton interactions that mediate dynamic cell adhesion[19].

Recently, we have shown that the above conformational changes in integrin αVβ3 can be prevented by binding of the αVβ3-specific peptide hFN10, a modified high-affinity mutant peptide derived from the 10th type III domain of fibronectin (FN10)[20]. We traced this unexpected effect of a ligand to a stable, key π–π stacking interaction between the indole derivative, $W^{1496}$, that immediately follows the RGD motif of hFN10, and the β3-$Y^{122}$ in the ligand binding βA domain of αVβ3[20], a conclusion supported in a subsequent molecular dynamics study[21]. In this study, we evaluate the applicability of building such π interactions for generation of pure orthosteric antagonists of other integrins, such as αIIbβ3. Accordingly, we have engineered a peptide and a small molecule targeting αIIbβ3 that are effective in preventing vascular thrombosis while preserving hemostasis, thus establishing a causal link between partial agonism and adverse outcomes, paving the way for potentially safer integrin-targeted medical therapies.

## Results

**Development of peptide Hr10.** Superimposing the βA domains from the crystal structures of αIIbβ3/eptifibatide complex (2vdn.pdb)[18] and αVβ3/hFN10 (4mmz.pdb; Fig. 1a) revealed a potential clash between hFN10 and αIIb propeller, involving $S^{1500}K$ in the C-terminal F-G loop of hFN10 and $V^{156}E$ in the longer helix-containing D2-A3 loop of αIIb. In addition, the short ligand

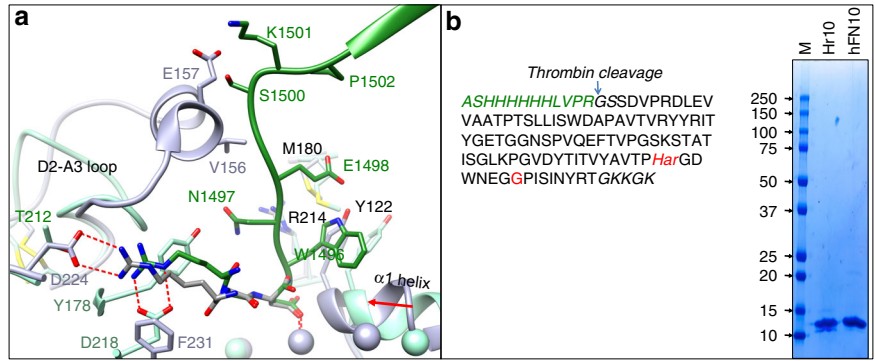

**Fig. 1 Structure-guided design and binding properties of Hr10. a** Ribbon diagrams of the crystal structures of αVβ3/hFN10 (light green, 4mmz.pdb) and αIIbβ3/eptifibatide (light purple, 2vdn.pdb) superposed on the respective βA domains, with the metal ions at the ligand-associated metal-binding site (LIMBS), MIDAS, and adjacent to MIDAS (ADMIDAS) shown as spheres in the respective colors. Relevant segments of the propeller and βA domains and of hFN10 (dark green) and eptifibatide (dark gray) are shown. The MIDAS ion is ligated by the aspartate residue from each ligand. Residues (single letter code) specific to each structure are shown in the respective color, with residues or loops common in both shown in black. Oxygen, nitrogen, and sulfur atoms are in red, blue, and yellow, respectively. The inward movement (red arrow) of the α1 helix and ADMIDAS ion in αIIbβ3, driven by binding of the partial agonist eptifibatide, is absent in hFN10-bound αVβ3, the result of a π–π interaction between hFN10-$W^{1496}$ and βA-$Y^{122}$. βA-$R^{214}$ and βA-$M^{180}$ contribute to the stability of hFN10-$W^{1496}$. Eptifibatide-Har[2] forms a bidentate salt bridge with αIIb-$D^{224}$, whereas hFN10-$R^{1493}$ contacts αV-$D^{218}$ (replaced by $F^{231}$ in αIIb). **b** The translated sequence of His-tagged Hr10 lacking the N-terminal methionine. Har and glycine substitutions are indicated in red. Alien residues are in italics (green text). The isotopically averaged calculated molecular weight of Hr10 was 11,969.3 (protein calculator v3.4 http://protcalc.sourceforge.net/cgi-bin/protcalc). Inset, Coomassie stain of a whole 10–20% SDS–PAGE showing purified Hr10 and hFN10 (8 μg in each lane). MW markers (in kDa) are indicated.

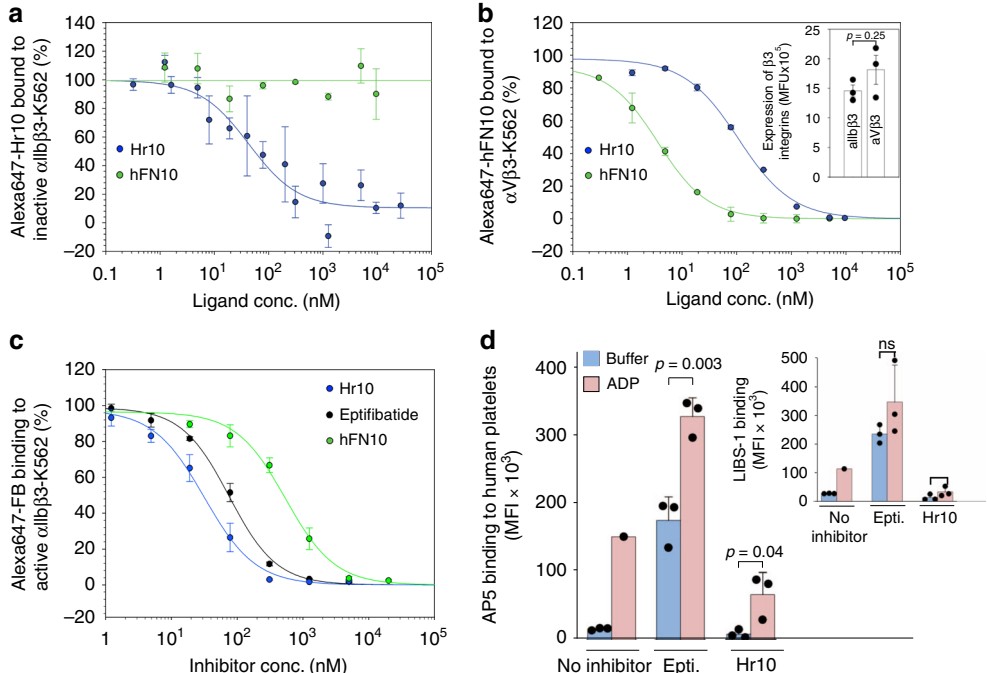

**Fig. 2 Binding properties of Hr10. a** Dose–response curves comparing displacement of Alexa647-labeled Hr10 from inactive αIIbβ3-K562 cells by unlabeled Hr10 ($n = 5$, 9 replicates) or hFN10 ($n = 3$) analyzed by fluorescence-activated cell sorting (FACS). The mean fluorescence intensity (MFI) values for individual experiments were initially fit with a binding curve to determine minimum and maximum MFI values to use in scaling the data. The blue and green lines in **a**–**c** are a least-squares fit to the averages. The points and error bars indicate the mean and standard error for the scaled data. **b** Dose–response curves comparing displacement of Alexa647-labeled hFN10 binding to αVβ3-K562 cells by unlabeled hFN10 or Hr10. Data were generated as in **a**. Inset, histograms (mean +/− s.d., $n = 3$ independent experiments) showing stable expression of the respective recombinant β3 integrin on K562 cells as determined by binding of the β3-specific mAb VI-PL2. Indicated $p$ value is determined by Student's $t$-test, two-tailed. **c** Dose–response curves (mean +/− s.d., $n = 3$ independent experiments) generated from FACS analysis showing displacement of Alexa-647 labeled fibrinogen (FB) bound to activated αIIbβ3-K562 in the presence of increasing concentrations of unlabeled Hr10, eptifibatide, or hFN10. The MFI values from the three separate FACS analyses were normalized individually before averaging as described in the methods. The IC50 values are stated in the text. Binding affinity of Hr10 to αIIbβ3 was significantly higher than that of eptifibatide ($p = 1.8 \times 10^{-5}$, $F$-test, two-tailed). **d** Histograms (mean + s.d., $n = 3$ independent experiments) showing effect of Hr10 vs. eptifibatide (each at 1.5 μM) on integrin conformational changes. Binding of the activation-sensitive mAb AP5 or the extension-sensitive mAb LIBS-1 (inset) to human platelets in the absence or presence of ADP (5 μM) was assessed following flow cytometry. Binding of the two conformation-sensitive mAbs to ADP-activated platelets in the absence of the inhibitors was often complicated by platelet aggregation, thus only single measurements could be obtained for each mAb in this case. Indicated $p$ values were determined by Student's $t$-test, two-tailed; ns, not significant.

$R^{1493}$ of hFN10 cannot make the critical bidentate salt bridge with αIIb-$D^{224}$. These feature account for the known inability of hFN10 to bind αIIbβ3[22]. We therefore substituted $S^{1500}K$ in hFN10 with glycine and replaced the ligand $R^{1493}$ with the longer L-homoarginine (Har; Fig. 1b), changes that we predicted would not adversely affect FN10 folding or the π–π stacking interaction between βA-$Y^{122}$ and $W^{1496}$ of the new peptide Hr10. The presence of Har in purified Hr10 (Fig. 1b, inset) was initially confirmed by mass spectroscopy (Supplementary Fig. 1).

**Hr10 is an RGD-based pure antagonist of αIIbβ3.** We measured binding of fluorescently labeled Hr10 and hFN10 to inactive recombinant αIIbβ3 and to αVβ3, each stably expressed in equivalent amounts on K562 cells (Fig. 2a, b; Supplementary Fig. 2). Hr10 bound αIIbβ3 with high affinity (IC50 = 58.8 +/− 24.1 nM; mean ± s.e.), with hFN10 showing no measurable binding to inactive αIIbβ3 (Fig. 2a). Hr10 continued to bind αVβ3-K562 but with much lower affinity when compared to hFN10 (IC50s of 107.9 +/− 23.1 nM and 3.6 +/− 0.72 nM, respectively; Fig. 2b). Hr10 inhibited binding of Alexa647-labeled soluble FB to preactivated αIIbβ3 with high affinity (IC50 30.3 ± 4.8 nM, mean ± s.e.) that was ~2.5-fold higher than that of eptifibatide (73.2 ± 7.0 nM, $p = 1.79 \times 10^{-5}$, $F$-test, two-tailed; Fig. 2c). hFN10 bound minimally to preactivated αIIbβ3-K562,

with an order of magnitude higher IC50 of 474.0 ± 73.4 nM (Fig. 2c).

Binding of eptifibatide at a clinically effective dose of 1.5 μM[23] to human platelets induced conformational changes in αIIbβ3 reported by binding of the activation-sensitive and extension-sensitive mAbs AP5 and LIBS-1, respectively[20] (Fig. 2d). These changes were markedly enhanced in presence of 5 μM ADP. In contrast, binding of Hr10 at 1.5 μM did not induce these changes directly, and also suppressed binding of AP5 and LIBS-1 to ADP-activated platelets (Fig. 2d). Thus Hr10 acts as a pure orthosteric antagonist of αIIbβ3.

**Crystal structure of αVβ3/Hr10 complex.** To elucidate the structural basis of pure antagonism, we determined the crystal structure of αVβ3/Hr10 complex at 3.1 Å resolution (Fig. 3a and Supplementary Table 1) by soaking Hr10 into preformed αVβ3 ectodomain crystals (crystal packing of the αIIbβ3 ectodomain does not allow access of large ligands such as Hr10 to the MIDAS). Hr10-$Har^{1493}$ forms a bidentate salt bridge with αV-$D^{218}$ and a cation–π interaction with αV-$Y^{178}$ but does not contact αV-$Thr^{212}$ (which replaces αIIb-$D^{224}$). Hr10-$D^{1495}$ directly coordinates the metal ion at MIDAS, with Hr10-$W^{1496}$ making a π–π stacking interaction with βA-$Y^{122}$, stabilized by an Hr10-$W^{1496}$ S–π interaction with βA-$M^{180}$ (Fig. 3a), and by a

**Fig. 3 Crystal structure of αVβ3/Hr10 complex. a** Ribbon diagram of the crystal structure of αVβ3-bound Hr10 (same view as Fig. 1a) showing 2fo–fc map at 1.0 σ (blue mesh) of the ligand-binding region. Relevant portions of Hr10 (light green), αV propeller (light blue), and the βA domain of β3 subunit (rose color) are shown. Side chains of amino acids (single letter code) are shown as sticks in the respective colors. The $Mn^{2+}$ ions at LIMBS, MIDAS, and ADMIDAS are in grey, cyan, and magenta spheres, respectively. Oxygen, nitrogen, and sulfur atoms are colored as in Fig. 1a. Water molecules are not shown. Hr10's $W^{1496}$ forms a π–π interaction with $βA-Y^{122}$, and Hr10-Har$^{1493}$ forms a bidentate salt bridge with αV-D$^{218}$. **b** Ribbon diagrams of the crystal structures of Hr10/αVβ3 (light green) and eptifibatide/αIIbβ3 (light purple, 2vdn.pdb) superposed on the βA domain of each. View, domain, side chain, and metal ion colors are as in **a**. Note the removal of the predicted clash of Hr10 with D2-A3 loop of αIIb and predicted formation of Hr10-Har$^{1493}$/αIIb-D$^{224}$ salt bridge.

hydrogen bond between the carbonyl oxygen of $W^{1496}$ and Nε of $βA-R^{214}$. Bound Hr10 prevented the activating inward movement of the α1 helix (reported by $β3-Y^{122}$) toward MIDAS, and the conformational changes at the C-terminal end of the βA domain that normally initiate integrin extension. Superposition of the βA domains from the αVβ3/Hr10 and αIIbβ3/eptifibatide structures (Fig. 3b) shows that the $S^{1500}K/G$ substitution removes the predicted clash with the αIIb propeller. The Nε, Nh1, and Nh2 amino groups of Hr10-Har$^{1493}$ superpose well on those of eptifibatide-Har$^2$ and could likewise form the critical bidentate salt bridge with αIIb-D$^{224}$, accounting for the observed high-affinity binding of Hr10 to αIIbβ3. $βA-Y^{122}$ is replaced with Phe$^{122}$ in mouse β3, and the stabilizing salt bridge $βA-R^{214}$ makes with $βA-D^{179}$ (Fig. 3a) is replaced with a H-bond with $βA-N^{179}$ in mouse, both substitutions likely contributing to the weak binding of Hr10 to mouse αIIbβ3.

**Effects of Hr10 on platelet aggregation and secretion.** Hr10 was as effective as eptifibatide in blocking collagen-induced platelet aggregation but was somewhat less effective in blocking ADP- or TRAP-induced aggregation (Fig. 4a–d). The adenine nucleotides ADP and ATP are coreleased from dense (δ)- granules during platelet activation and interact with platelet $P_2$ receptors to amplify ongoing platelet activation. Both Hr10 and eptifibatide (at 1.5 μM) significantly inhibited ADP (20 μM)-induced ATP secretion from δ- granules (Fig. 4e), but did not significantly suppress ADP-induced release from α-granule (reported by CD62P surface expression) or lysosomes (reported by CD63 surface expression; Fig. 4f). This observation has been previously documented for αIIbβ3 antagonists[24] and lead to the suggestion that ADP-induced α-granule release can occur independently of αIIbβ3[25].

**Hr10 preserves thrombin-induced clot retraction.** Clot retraction normally helps secure hemostasis in vivo as evidenced by increased bleeding in mice with impaired clot retraction[26], or in recipients of any of the three anti-αIIbβ3 drugs[4,27,28]. We compared the effects of Hr10 and eptifibatide on thrombin-induced clot retraction in fresh human platelet-rich plasma (PRP)[29]. The kinetics of clot retraction were determined from quantification of serial images of the reaction acquired every 15 min for the 2-h duration of the assay. As shown in Fig. 5a, b, Hr10 (at 1.5 μM) did not inhibit clot retraction vs. buffer alone ($p = 0.125$, F-test,

two-tailed). In contrast, eptifibatide (at 1.5 μM) significantly blocked clot retraction vs. buffer ($p = 4.5 \times 10^{-5}$, F-test, two-tailed), as previously shown[27,30]. αIIbβ3 antagonists that block platelet aggregation but not clot retraction have been reported to exhibit affinities to inactive αIIbβ3 that are 2–3 logs lower than those to active αIIbβ3[3]. This was not the case with Hr10, however: its binding affinities to inactive ($IC_{50} = 58.8 +/- 24.1$ nM, nine determinations from five experiments) and active αIIbβ3 ($IC_{50}$ $35.2 +/- 5.7$ nM, $n = 3$ experiments) were not significantly different ($p = 0.54$, F-test, two-tailed; Fig. 5c), and were comparable to the affinity of eptifibatide to αIIbβ3 on resting platelets ($k_D = 120$ nM)[31].

**Effects of Hr10 in humanized mice.** To evaluate the effects of the peptides Hr10 and eptifibatide on nascent thrombus formation under flow, we induced thrombin-mediated arteriolar injury in a humanized mouse model shown to predict clinical efficacy of anti-platelet agents[32]. NSG (NOD-scid-IL-2Rγ$^{null}$) mice were made homozygous for human von Willebrand factor R$^{1326}$H (vWF$^{RH/RH}$)[32], a substitution that switches binding of vWF from mouse to human glycoprotein (GP) Ib/IX, accounting for the increased bleeding risk in these mice unless infused with human platelets. To assess the effects of Hr10 and eptifibatide on thrombus formation, each inhibitor was given to mice preinfused with human platelets (eptifibatide[32], like Hr10, binds poorly to mouse αIIbβ3). Hr10 in equimolar concentrations to eptifibatide was as effective in preventing nascent occlusive thrombus formation at multiple sites of laser-induced arteriolar injury in the cremaster muscle (Fig. 6a). Hr10 did not cause significant human platelet clearance (Supplementary Fig. 3). Importantly, and in contrast to eptifibatide, Hr10 did not cause significant loss of blood in these mice (Fig. 6b).

**Converting tirofiban into a pure αIIbβ3 antagonist.** We next explored the feasibility of converting the αIIbβ3-specific non-peptidic partial agonist drug tirofiban (molecular weight of 495.08; Fig. 7a, left panel) into a pure antagonist, guided by the present crystal structure of αVβ3/Hr10 complex. Superposing the βA domains of αVβ3/Hr10 and αIIbβ3/tirofiban (2vdm.pdb) structures show that the acidic moiety of each ligand and the following amide are nearly superimposable (r.m.s.d. = 0.9648; Fig. 7b), suggesting that replacing the butane-sulfonamide moiety of tirofiban with an indole group could create the critical π–π

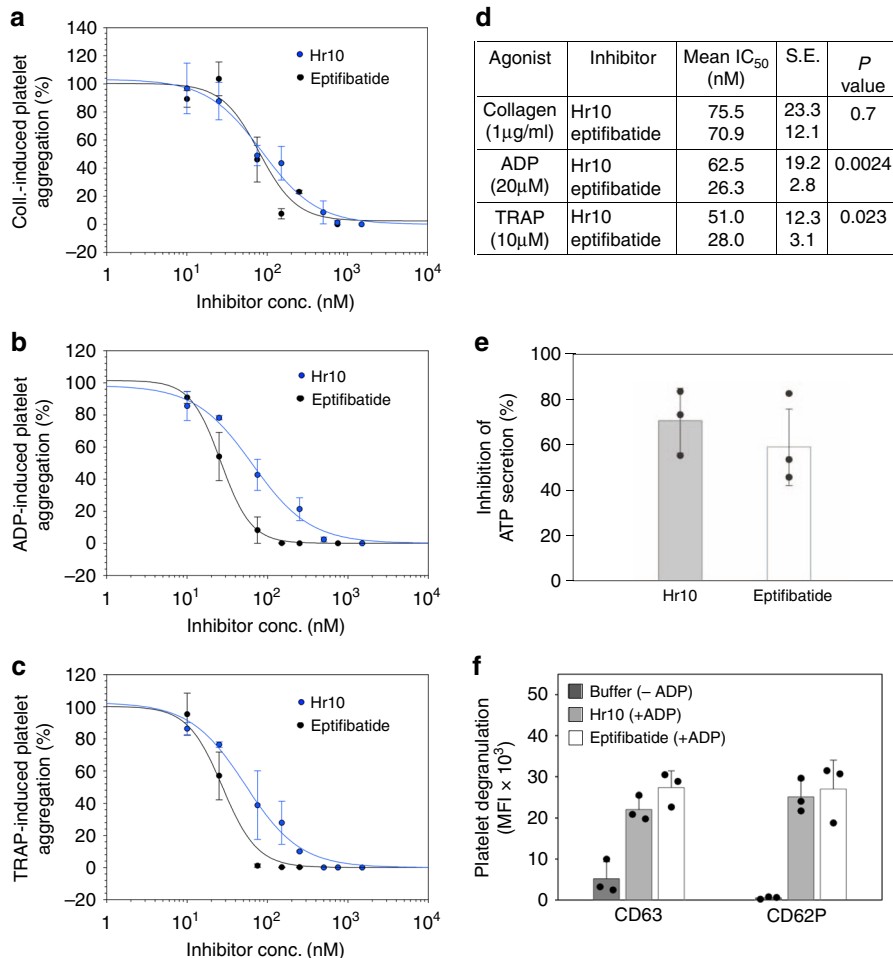

**Fig. 4 Effect of Hr10 and eptifibatide on platelet aggregation and secretion. a–c** Dose–response curves (mean $+/-$ s.e., $n = 3$ experiments, each from a different donor) showing effects of the inhibitors on aggregation induced by collagen (Coll.) (2 μg/ml) **a**, ADP (20 μM) **b**, or TRAP (10 μM) **c**. Points for the integrated impedance from the three experiments were individually normalized prior to averaging and are displayed with least-squares fits to the mean values. The respective IC$_{50}$, s.e., and p values (F-test, two-tailed) are listed in **d**. **e–f** Histograms (mean + or $+/-$ s.d., $n = 3$ independent experiments) showing the effect of Hr10 and eptifibatide (each at 1.5 μM) on ADP (20 μM)-induced ATP secretion (**e**, $p = 0.41$, Student's t-test, two-tailed) and surface expression of CD63 and CD62P **f** on human platelets. No differences in expression of CD63 ($p = 0.15$) or CD62P ($p = 0.72$), both Student's t-test, two-tailed, were found in platelets exposed to eptifibatide or Hr10. Buffer sample in **f** represents marker expression in the absence of ADP.

stacking interaction with βA-Y$^{122}$. We selected the indole derivative benzoxazole in order to stabilize this interaction further by formation of a hydrogen bond between the benzoxazole oxygen and Nε of βA-R$^{214}$ as in the αVβ3/Hr10 structure. Structure of such modified tirofiban (M-tirofiban; Fig. 7a, right panel; Supplementary Fig. 4) in complex with inactive αIIbβ3 (3fcs.pdb) was then modeled in Coot[33] by geometry minimization with a library generated by eLBOW in Phenix[34]. In this model (Fig. 7b), the RGD-like moiety of M-tirofiban superimposes nicely onto that of tirofiban, with the benzoxazole moiety forming a π–π stacking interaction (4.4 Å) with βA-Y$^{122}$, and the benzoxazole oxygen forming a hydrogen bond (3.2 Å) with Nε of βA-R$^{214}$, arrangements predicted to freeze αIIbβ3 in the inactive conformation.

**In vitro and in vivo activities of M-tirofiban**. The αIIbβ3-specific M-tirofiban (Supplementary Fig. 5) effectively blocked FB binding to preactivated αIIbβ3 (Fig. 7c), and prevented ADP-induced human platelet aggregation even better than eptifibatide (Fig.7d). The observed ~10-fold reduction in affinity of M-tirofiban vs. tirofiban to αIIbβ3 likely reflects weaker H-bonding of M-tirofiban's benzoxazole oxygen (vs. tirofiban's sulfonamide oxygen) with Nε of β3-R$^{214}$ and perhaps loss of

hydrophobic contacts with the integrin by the deleted butane moiety.

In contrast to tirofiban (used at a clinically effective concentration of 0.15 μM[23]), M-tirofiban (at the equipotent concentration of 1.5 μM) neither induced the activating conformational changes in αIIbβ3 directly nor when these changes were induced by ADP (Fig. 8a). It also preserved thrombin-induced clot retraction vs. tirofiban (Fig. 8b), and exhibited equivalent binding affinities to active and inactive αIIbβ3 (Fig. 8c, d). In vivo, M-tirofiban did not affect human platelet clearance (Supplementary Fig. 3), and was as effective as tirofiban and Hr10 in preventing redox-induced thrombosis following carotid artery injury (Fig. 9a). Significantly, like Hr10 and in contrast to tirofiban, M-tirofiban did not increase blood loss (Fig. 9b) or bleeding time (Fig. 9c) in mice.

## Discussion

The Achilles heel of current anti-thrombosis drugs that directly target αIIbβ3 is serious bleeding, an adverse outcome that remains high with use of the newer inhibitors of P$_2$Y$_{12}$ and thrombin receptors[35,36]. Several attempts are being made to develop new anti-thrombosis drugs that maintain efficacy but

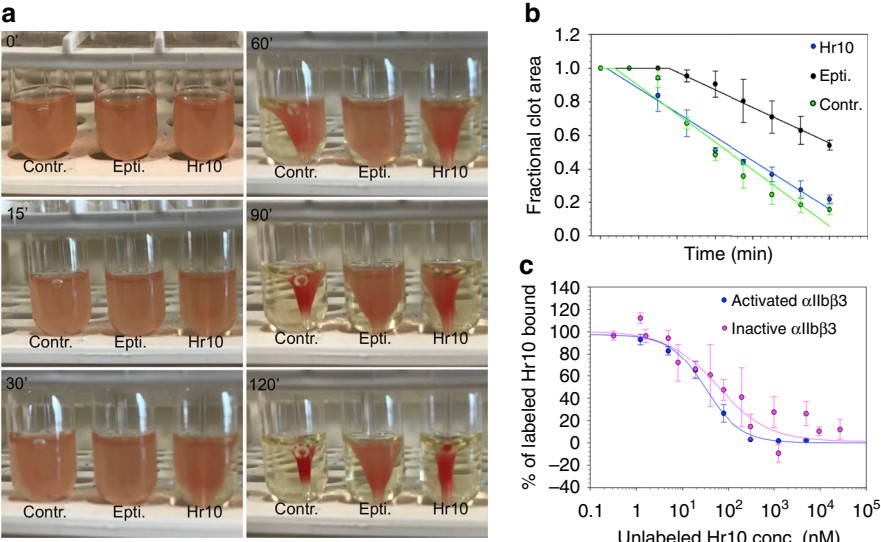

**Fig. 5 Clot retraction and integrin binding studies. a** Kinetics of clot retraction in the absence (Contr.) and presence of Hr10 or eptifibatide (Epti.) from a representative experiment, one of three conducted. Clot retraction took place around a central glass rod. A total of 5 μl of red blood cells were added per 1 ml reaction to enhance color contrast for photography. Photographs shown were taken at timed intervals (in minutes) after addition of thrombin. **b** Time course (mean +/− s.e.) from three clot retraction experiments, including the one shown in Fig. 5a. The plot shows the fractional area occupied by the clot at 15-min intervals with a linear regression through the points. No significant differences ($p = 0.125$, F-test, two-tailed) were found in kinetics of clot retraction in buffer vs. Hr10. A lag period is noted with eptifibatide and clot retraction was significantly reduced vs. buffer ($p = 4.5 \times 10^{-15}$, F-test, two-tailed). **c** Dose–response curves comparing displacement of Alexa488-labeled Hr10 binding to PT-25-activated αIIbβ3 on K562 cells by increasing concentrations of unlabeled Hr10 with that to the inactive receptor (Fig. 2a). Cell binding was analyzed by FACS. The mean fluorescence intensity (MFI) values of binding to active αIIbβ3 (mean +/− s.e. of three independent experiments) were initially fit with a binding curve to determine minimum and maximum MFI values to use in scaling the data. The points and error bars indicate the mean and standard error for the scaled data. The red and black lines are a least-squares fit to the averages. No significant differences were found ($p = 0.54$, F-test, two-tailed).

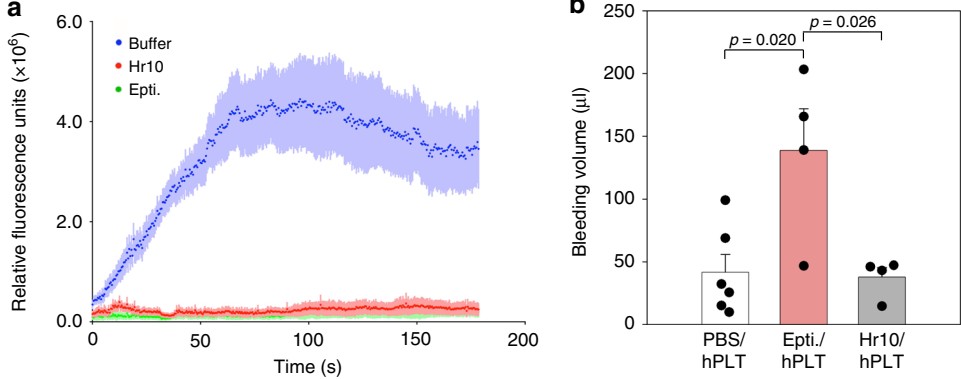

**Fig. 6 Hr10 inhibits thrombosis without causing bleeding. a** Graphs showing kinetics of human platelet accumulation at nascent laser-induced injury sites in cremaster muscle, 5 min following infusion of PBS or equimolar amounts of Hr10 or eptifibatide in the same buffer. Values represent the mean + s.e. of four mice per arm). There was no significant difference in human platelet accumulation in thrombi between Hr10- and eptifibatide-treated mice at each time point. **b** Histograms (mean + s.e.) showing baseline bleeding volume in vWF$^{RH/RH}$NSG mice infused with human platelets before (PBS; $n = 6$) or after administration of eptifibatide (Epti.; $n = 4$), or Hr10 ($n = 4$). Epti. caused excessive loss of blood (~10% of blood volume of a normal mouse), which was absent in Hr10-treated mice. Difference between PBS- and Hr10-treated mice was not significant ($p = 0.94$, Student's t-test, two-tailed). The other $p$ values are shown.

preserve hemostasis. These include targeting collagen receptors α2β1 and GPVI[37,38], accelerating ADP degradation with CD39[39] or interfering with ADP-induced cell signaling with a PI3Kβ inhibitor[40]. However, these approaches do not affect platelet activation induced by other potent agonists, and some targets (e.g., PI3Kβ and α2β1) are not platelet specific. Platelet–leukocyte interactions are also being targeted: interfering with binding of leukocyte integrin CD11b to platelet GP1bα delayed thrombosis without prolonging bleeding time in normal mice[41]. However, platelet–leukocyte interactions are mediated by multiple receptor–counterreceptor pairs, the relative importance of which may vary with the nature of the pathologic state. Two recent attempts targeted αIIbβ3 more directly. In one approach, a short cytoplasmic β3-derived peptide inhibited αIIbβ3 outside-in signaling and prevented thrombosis without prolonging bleeding time; however, it is not β3-integrin specific[30]. The second

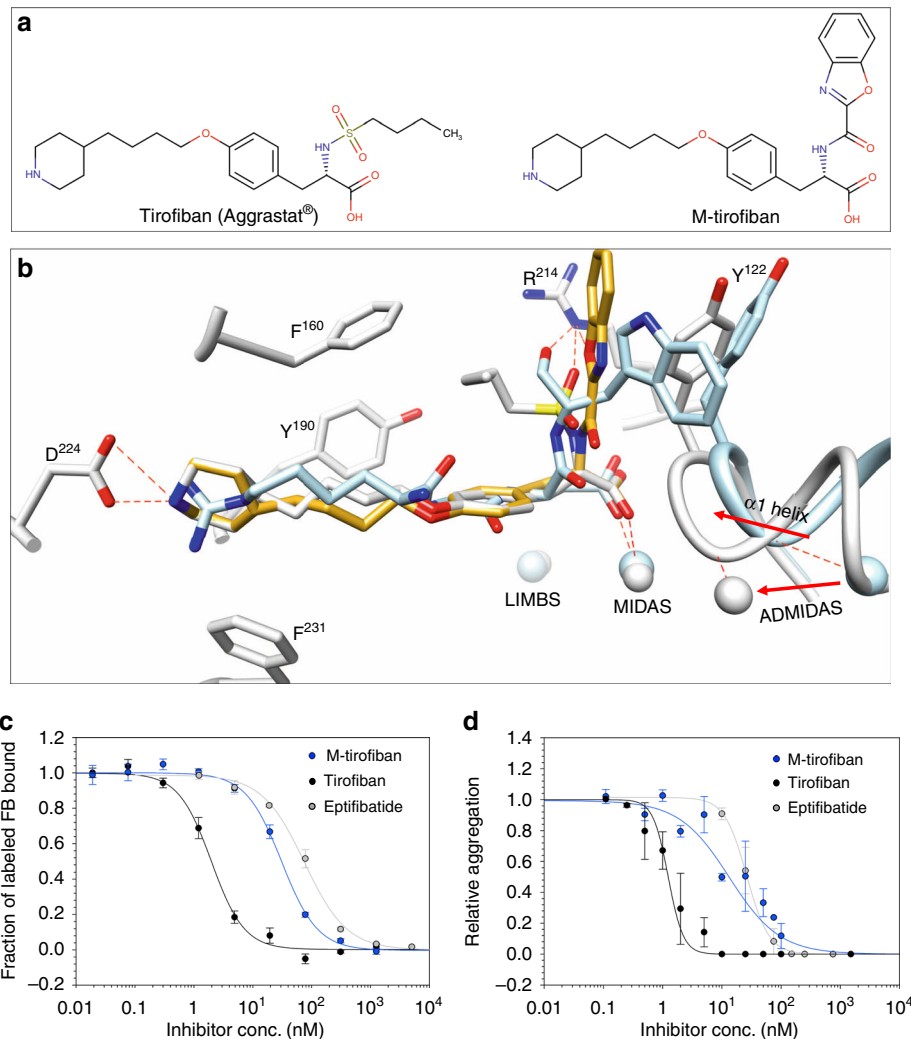

**Fig. 7 Engineering and in vitro activities of M-tirofiban. a** Chemical structure of tirofiban and M-tirofiban (see also Supplemental Fig. 4). **b** A model of bound M-tirofiban (gold) superposed on the structures of tirofiban/αIIbβ3 (gray, 2vdm.pdb) and the present αVβ3/Hr10 structure (light blue). The βA domain of each was used in superposition. The metal ions at LIMBS, MIDAS, and ADMIDAS and relevant residues are shown in the respective colors. Contacts are shown as dotted red lines. See text for details. **c** Dose–response curves (mean +/− s.e., $n = 4$ experiments) showing displacement of labeled FB bound to preactivated αIIbβ3-K562 by tirofiban or M-tirofiban, yielding IC$_{50}$s of 1.98 +/− 0.19 nM and 30.9 +/− 3.3 nM, respectively. Displacement of labeled FB by eptifibatide (73.2 ± 7.0 nM; Fig. 2c) is added for comparison. **d** Dose–response curves (mean +/− s.e.) showing effects of tirofiban and M-tirofiban on human platelet aggregation from three different donors (three tirofiban and five M-tirofiban determinations) induced by ADP (20 μM), yielding IC$_{50}$s of 1.41 +/− 0.23 nM and 18.5 +/− 5.4 nM, respectively, in comparison with eptifibatide (26.3 +/− 2.8 nM; Fig. 4b).

approach utilized low-affinity non-RGD small molecules that primarily engage the arginine pocket in αIIb[42]. These prevented FeCl$_3$-induced thrombotic arterial occlusion in mice but its effects on clot retraction or bleeding were not reported[43].

The present data show that the pure RGD-based αIIbβ3 antagonists Hr10 and M-tirofiban prevented thrombotic arterial and microvascular occlusion, and preserved hemostasis in humanized mice, thus demonstrating that partial agonism and antagonism of integrins are not inseparable. Pure orthosteric antagonism of αIIbβ3 offers significant advantages over the other approaches aimed at preserving hemostasis. First, by targeting the RGD-binding pocket of αIIbβ3 directly, such antagonists block binding of several prothrombotic ligands, some of which, such as CD40L[44], also bind leukocyte CD11b[45] and thus contribute to platelet–leukocyte interactions. Second, these high-affinity pure orthosteric inhibitors do not induce the conformational changes directly and block these changes when induced by inside-out integrin activation. Third, Hr10, a minor variant of a human natural ligand, is expected to be minimally immunogenic.

Whether certain preformed antibodies that recognize αIIbβ3 in complex with tirofiban[15] no longer do so when αIIbβ3 is bound by M-tirofiban or Hr10 can now be tested. Success in converting the partial agonist tirofiban into a pure antagonist using the present αVβ3/Hr10 structure also underscores the primacy of the stable π–π ligand-W[1496]/βA-Y[122] stacking interaction in preventing the activating global conformational change in αIIbβ3, and suggests that this approach may be applicable to engineering drug candidates targeting other integrins, where inadvertent conformational changes may also compromise patient safety.

The precise mechanism by which prevention of the agonist-induced conformational changes in αIIbβ3 by these pure orthosteric antagonists results in preservation of clot retraction is presently unknown, but there are several possibilities. Clot retraction occurs in response to the binding of polymeric fibrin to αIIbβ3, thus linking the integrin to actomyosin[46]. When compared with FB, polymeric fibrin binds αIIbβ3 with higher affinity[47]. So one possibility is that high affinity of the partial agonists is necessary to block fibrin–αIIbβ3 interaction and hence

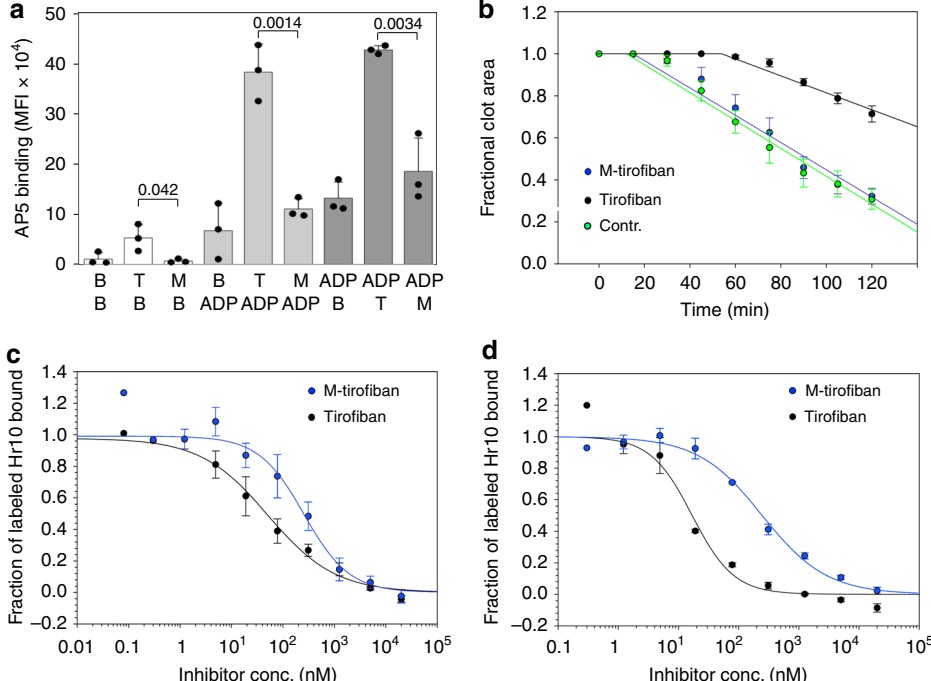

**Fig. 8 M-tirofiban is a pure αIIbβ3 antagonist. a** Histograms (mean + s.d., $n = 3$ independent experiments) showing binding of mAb AP5 to human platelets in presence of buffer (B), tirofiban (T; 150 nM), and M-tirofiban (M; 1.5 μM; white histograms), and before (gray histograms) or after (dark gray histograms) addition of ADP (5 μM). Numbers represent $p$ values (Student's $t$-test, two-tailed). No significant differences were found between buffer and M-tirofiban before ($p = 0.273$) or after ($p = 0.81$; both Student's $t$-test, two-tailed) ADP addition. **b** Kinetics of clot retraction in absence or presence of tirofiban, or M-tirofiban (mean +/− s.e., five experiments with three donors). Kinetics of clot retraction was not different between buffer control (contr.) and M-tirofiban ($p = 0.61$, $F$-test, two-tailed). **c, d** Dose–response curves (mean +/− s.e.) comparing displacement of Alexa488-labeled Hr10 binding to inactive (**c**, $n = 5$) or mAb PT-25-activated αIIbβ3- K562 (**d**, $n = 3$) by increasing concentrations of tirofiban or M-tirofiban in presence of physiologic concentrations of $Mg^{2+}$ and $Ca^{2+}$ (1 mM each). The respective $IC_{50}$s were 51.3 +/− 19.2 nM and 257.2 +/− 88.0 nM for inactive αIIbβ3, and 16.9 +/− 2.4 nM and 247.1 +/− 29.3 nM for active αIIbβ3. The lower affinities of both compounds are explained by the requirement for more inhibitor to displace high affinity binding of Hr10 (compared to binding of FB in Fig. 7c) to αIIbβ3.

clot retraction. This scenario is unlikely since Hr10 and eptifibatide have comparable affinities in blocking soluble FB binding to activated αIIbβ3 and in agonist-induced platelet aggregation. Preservation of clot retraction by the pure orthosteric antagonists could not be explained by a weaker affinity to inactive αIIbβ3[48], since affinities of the pure antagonists to inactive and active αIIbβ3 were similar. A recent study showed that fibrin binds αIIbβ3 even when all the RGD motifs in fibrin are deleted[47], suggesting the presence of MIDAS-independent fibrin-binding sites[49]. Since αIIbβ3 on non-activated platelets binds surface-immobilized fibrin[50,51], it may also do so when occupied by Hr10 or M-tirofiban. It has been shown that αIIbβ3-dependent fibrin clot retraction kinetics correlates with intracellular protein tyrosine dephosphorylation, which is inhibited by binding of eptifibatide or abciximab to αIIbβ3[27]. The availability of the pure orthosteric inhibitors of αIIbβ3 should now provide a new tool to further dissect the mechanisms linking integrin conformation to clot retraction.

The dual specificity of Hr10 to both β3 integrins is shared with the drug abciximab[52], a property thought to contribute to the long-term clinical benefits of abciximab in acute coronary syndrome[53,54]. In addition, dual specificity of abciximab to both β3 integrins has shown a wide range of anticancer effects (reviewed in ref. [55]). For example, abciximab was effective at blocking tumor growth and angiogenesis through targeting the interaction of tumor cells with platelets and endothelial cells, in addition to direct effects on the tumor tissue[56–59]. Hr10 may thus offer an attractive clinical candidate in this case with minimal immunogenicity or bleeding risk.

## Methods

**Reagents and antibodies**. Restriction and modification enzymes were obtained from New England Biolabs Inc. (Beverly, MA). Cell culture reagents were purchased from Invitrogen (San Diego, CA) or Fisher Scientific (Hampton, NH). The Fab fragment of mAb AP5[60] was prepared by papain digestion followed by anion exchange and size-exclusion chromatography. Hybridoma producing the β3 conformation-insensitive mAb AP3 was bought from ATCC (catalogue #ATCC HB®-242) and antibody purified by affinity chromatography. Alexa Fluor 488-conjugated mAbs against human CD62P (catalogue #sc-8419) and CD63 (catalogue#sc-5275) were bought from Santa Cruz Biotechnology, Dallas, TX. Alexa Fluor647-conjugated anti-human CD42b mAb (catalogue #FAB4067R) was from R&D Systems, Minneapolis, MN. APC-labeled goat anti-mouse Fc-specific antibody (catalogue #115-136-071) was from Jackson ImmunoResearch (West Grove, PA). Alexa Fluor-488-labeled F(ab')₂ fragment of mouse anti-human CD41a (catalogue #555465) and Alexa Fluor-647 rat anti-mouse CD41a F(ab')₂ (catalogue #624101) were from BD Biosciences. The activating mAb PT-25-2 was a gift from Dr. Makoto Handa[61]. Eptifibatide and tirofiban were purchased from Millipore-Sigma (Burlington, MA). The plasmid pCDF5-Har, containing two copies of a UAG recognizing tRNA and the tRNA synthase (Har-Rs) for charging UAG tRNAs with Har, and the *Escherichia coli* strain B-95ΔA containing a deletion of release factor 1 (*prfA*) and 95 synonymous TAG stop codon mutations, were kindly provided by Dr. Kensaku Sakamoto (RIKEN, Yokohama, Japan)[62]. L-Har and TRAP-6 were purchased from Bachem Americas, Inc. (Torrance, CA). ADP, collagen, ATP, Chrono-luminescence reagent, and human thrombin were purchased from Chrono-log (Havertown, PA).

**Characterization of M-tirofiban**. M-tirofiban was synthesized at the Organic Chemistry Collaborative Center, Columbia University Irving Medical Center, NY (Supplementary Fig. 4) using standard techniques. Its purity was ≥95% as determined by liquid chromatography–mass spectrometry performed on two different instruments, a Shimadzu 2010A and a Shimadzu 2020 UFLC mass spectrometer at wavelengths 220 and 254 nm, using a Waters Sunfire column (C18, 5 μm, 2.1 mm × 50 mm, a linear gradient from 5 to 100% B over 15 min, then 100% B for 2 min (A = 0.1% formic acid + $H_2O$, B = 0.1% formic acid + $CH_3CN$), flow rate 0.2000 ml/min). ¹H NMR and ¹³C NMR spectra were recorded on an

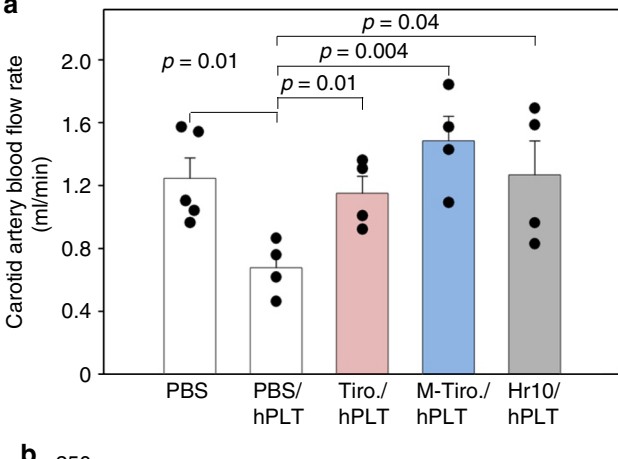

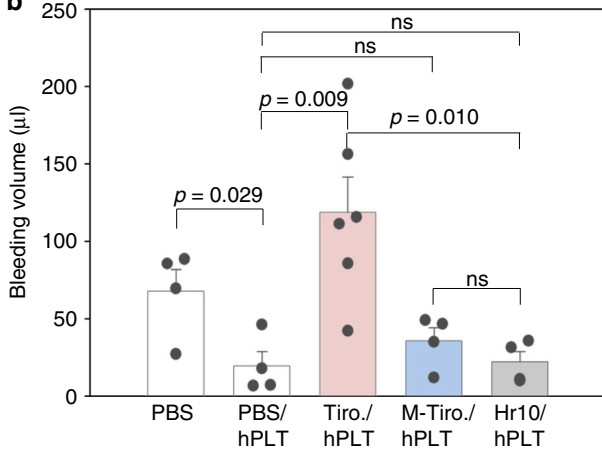

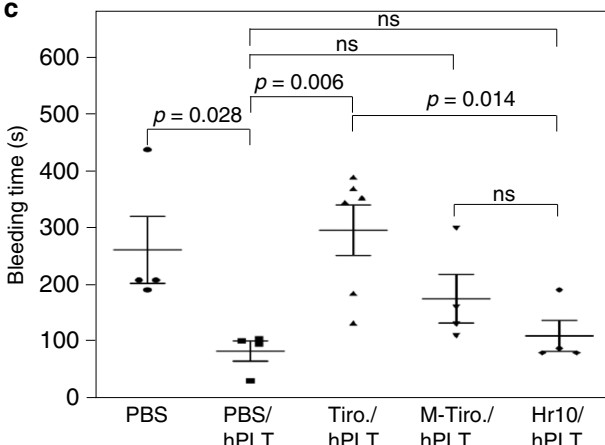

**Fig. 9 In vivo activities of M-tirofiban. a** Histograms (mean + s.e.) showing carotid artery blood flow over the 40-min observation period following redox-induced arterial injury. vWF$^{RH/RH}$ NSG male mice were either uninfused (PBS; $n = 5$) or infused with human platelets (hPLT) plus PBS ($n = 4$), tirofiban (Tiro.; $n = 4$), M-tirofiban (M-Tiro.; $n = 4$), or Hr10 ($n = 4$). **b** Blood loss (mean + s.e.) in vWF$^{RH/RH}$NSG male mice uninfused (PBS; $n = 4$) or infused with hPLT plus PBS ($n = 4$), tirofiban (Tiro.; $n = 6$), M-tirofiban (M-Tiro.; $n = 4$), or Hr10 ($n = 4$). $p$ values are indicated. **c** Bleeding time analysis (mean +/− s.e.) in vWF$^{RH/RH}$ NSG male mice treated as in Fig. 6b. Statistical tests by Student's $t$-test, two-tailed; ns, not significant.

Agilent 400-MR 400-MHz NMR spectrometer, operating at 400 MHz ($^1$H NMR) and 101 MHz ($^{13}$C NMR). Chemical shifts are given in units δ (ppm). The proton resonance of residual solvent (δ 3.31 for $d_4$-methanol) was used as the internal reference for $^1$H NMR spectra. The chemical shifts of $^{13}$C spectra are reported relative to the carbon signal of $d_4$-methanol at δ 49.00. All reagents and solvents

were used as received from major commercial suppliers, such as Sigma-Aldrich, Fisher Scientific, and Alfa Aesar without further purification. All air- or moisture-sensitive reactions were run under an atmosphere of argon in oven-dried glassware.

**Plasmids, mutagenesis, protein expression, and purification.** Human αVβ3 ectodomain and hFN10 were expressed in insect cells and BL21-DE3 bacteria, respectively, and each purified by affinity chromatography followed by gel filtration[20]. Hr10 was generated by replacing S$^{1500}$K in hFN10 with a glycine residue and introducing a TAG stop codon at position 1493 by PCR-based mutagenesis with the Quick-change kit (Agilent Technologies). Hr10 cDNA was cloned into the bacterial expression plasmid pET11a and verified by DNA sequencing. A bacterial stock of *E. coli* strain B-95ΔA containing plasmids pCDF5-Har and pET-11a/Hr10 grown in LB media supplemented with 5 mM L-Har, 50 μg/ml kanamycin (pCDF5-Har), and 100 μg/ml ampicillin (pET-11a) was prepared and used to express Hr10 protein. Bacterial cultures at ~0.5 absorbance (600 nm) were induced with 0.3 mM IPTG and grown for 8 h at room temperature (RT). Hr10 was purified as for hFN10 and purity assessed by fractionation on gradient sodium dodecyl sulfate–polyacrylamide gel electrophoresis (SDS–PAGE) gels followed by Coomassie staining.

**Cell lines, cell culture, and transfection.** Human K562 cells stably expressing αVβ3 (αVβ3-K562) have been described[20]. K562 cells stably expressing αIIbβ3 (αIIbβ3-K562) were kindly provided by Dr. Jennifer Cochran (Stanford University, CA)[63]. Cultured K562 were maintained in Iscove's modified Dulbecco's medium plus G418 (0.5–1.0 mg/ml), supplemented with 10% fetal calf serum, 2 mM L-glutamine, and penicillin and streptomycin.

**Ligand binding and flow cytometry.** For ligand binding assays, αIIbβ3-K562 or αVβ3-K562 cells ($1 \times 10^6$) were suspended in 100 μl of buffer A (20 mM Hepes, 150 mM NaCl, pH 7.4, containing 0.1% [w/v] bovine serum albumin, and 1 mM each of MgCl$_2$ and CaCl$_2$) and incubated first with Alexa-647-labeled Hr10 or hFN10 (each at 3–10 μg/ml) for 30 min at RT. For competition studies, 100 μl of mAb PT-25-activated αIIbβ3-K562 ($1 \times 10^6$) were incubated for 30 min at RT with serially diluted concentrations of unlabeled Hr10, hFN10, or eptifibatide in the presence of 0.5 μM Alexa647-conjugated FB. Cells were washed with 4 ml of buffer A and centrifuged for 5 min at 525×$g$, resuspended, fixed in 2% paraformaldehyde, and analyzed using FACSCalibur or BD-LSRII flow cytometers (BD Biosciences). Ligand binding was expressed as mean fluorescence intensity (MFI), as determined using FlowJo software. Mean and s.d. from independent experiments were calculated and compared using Student's $t$-test.

**Platelet aggregation and ATP secretion.** Platelet aggregation and ATP secretion in whole blood were measured using a Chrono-Log model 700 two-channel lumi-aggregation system following the manufacturer's instructions. Human blood samples were obtained from healthy volunteers, who provided written informed consent under a study that complied with all relevant ethical regulations and approved by the Human Subjects Committee at the Massachusetts General Hospital in accord with the Helsinki Principles. Blood was drawn directly into 3.2% sodium citrate and used within 3 h. None of the subjects were taking any medications for at least 10 days prior. For impedance aggregation measurements, 0.5 ml of whole blood was mixed with 0.5 ml physiologic saline supplemented with inhibitors and incubated at 37 °C for 5 min without stirring. Measurements were performed with stirring at 1200 rpm at 37 °C. Values for each data point represent impedance measurements following application of agonist, integrated over 5 min. Data points for an individual dose curve were serially collected from a single draw and analyzed with SigmaPlot (Systat Software, San Jose, CA) using a least-square fit to a logistic curve, and the IC$_{50}$ values determined from the fitted parameter. ATP secretion proceeded similarly except that 0.45 ml of whole blood were added to 0.45 ml of saline supplemented with various concentrations of Hr10 or eptifibatide to produce the desired concentration in 1.0 ml. Following incubation for 5 min at 37 °C, 100 μl of Chrono-lume reagent was added and aggregation initiated. The luminescence signal was quantified with a non-aggregated sample supplemented with an ATP standard.

**Binding of mAbs.** αVβ3- or αIIbβ3-K562 cells ($0.5 \times 10^6$ in 100 μl of buffer A) were incubated in the absence or presence of unlabeled Hr10 or eptifibatide, each at 1.5 μM, for 20 min at RT. Alexa647-labeled AP5 Fab or unlabeled LIBS-1 mAb (each to 10 μg/ml) were added, and cells incubated for an additional 30 min before washing with 4 ml of buffer A and centrifugation for 5 min at 525×$g$. APC-labeled goat anti-mouse Fc-specific antibody (at 10 μg/ml) was added to LIBS-1-bound cells for an additional 30 min at 4 °C. Afterward, cells were washed and processed for flow cytometry. To quantify ADP-induced expression of CD62P and CD63 on human platelets, ADP (at 20 μM) was added to 100 μl of 3.2% sodium citrate whole blood that was preincubated with Hr10 or eptifibatide (at 1.5 μM each) for 5 min at RT. Untreated anticoagulated whole blood served as negative control. Alexa488-labeled anti-CD62P or anti-CD63 mAbs (at 10 μg/ml) plus Alexa647-labeled anti-CD42b mAb (at 10 μg/ml) were then added to the blood samples for 20 min at RT. Cells were washed, fixed in 2% paraformaldehyde, and CD62P and CD63 expression analyzed by flow cytometry in the CD42b-positive

population. Mean and s.d. from three independent experiments were compared using Student's *t*-test.

**Crystallography, structure determination, and refinement.** Human αVβ3 ectodomain was purified and crystallized by the hanging drop method as previously described[16]. Hr10 was soaked for 3 weeks into the preformed αVβ3 crystals at 1.5 mM in the crystallization well solution containing 1 mM $Mn^{2+}$. Crystals were harvested in 12% PEG 3500 in 100 mM sodium acetate, pH 4.5, 800 mM NaCl plus 1 mM $Mn^{2+}$, cryoprotected by addition of glycerol in 2% increments up to 24% final concentration, and then flash-frozen in liquid nitrogen. Diffraction data were collected at ID-19 of APS, indexed, integrated, scaled by HKL2000[64], and solved by molecular replacement in PHASER using 3ije.pdb and 1fnf.pdb as the search model. The structure was refined with Phenix using default restraints, positional and individual temperature factor, and translation-liberation-screw. Automatic optimization of X-ray and stereochemistry, and Ramachandran restriction were used in the final cycle. Data collection and refinement statistics are shown in Supplementary Table 1. The coordinates and structure factors of αVβ3/Hr10 have been deposited in the Protein Data Bank under accession code 6NAJ. Structural illustrations were prepared with Chimera.

**Generation of vWF^R1326H knock-in NSG mice.** CRISPR/Cas 9 technology was used to generate the vWF^R1326H knock-in (KI) mice of NSG background with a mutation of specific nucleotide at the exon 28 of the mouse vWF gene, resulting in replacing the arginine (codon CGT) at amino acid no. 1326 by histidine (codon CAT). An sgRNA was designed according to the online resources, the sgRNA Designer: CRISPRko and the Cas-OFFinder, and the sgRNAs with less than three mismatches and less than 25 off-target sites were used. The sgRNA target sequence was 5′- CTTGAGCTCAA GGTAGGCAC-3′. The histidine codon was repaired into the gene with a single-stranded oligo. (5′-ACATCTCTCAGAAGCGCATCCG CGTGGCAGTGGTAGAGTACCATGATGGATCCCATGCTTATCTTGAGCTC AAGGCCCGGAAGCGACCCTCAGAGCTTCGGCGCATCACCAGCCAGA TTA-3′(Integrated DNA technologies, Inc.). Preparation of sgRNA and Cas9 RNA for pronucleus microinjection followed the instructor's manual (AmpliCap-MaxTM T7 High Yield Message Maker kit). Pronuclear microinjection was performed on fertilized eggs from NSG mice. Genotyping of founder mice was performed by PCR, TA-cloning, followed by Sanger DNA sequencing. The primer sequences for PCR genotyping were 5′-TCACTGTGATG GTGTGAACC-3′ pairing with 5′-CTGACTATCTC ATCTCTTC-3′. PCR condition was 95 °C, 5 min, followed by 35 cycles of 95 °C, 30 s; 55 °C, 30 s; and 72 °C, 30 s, and a final extension at 72 °C, 7 min. TA-cloning followed the instructor's manual (T3 Cloning kit; ZGene Biotech Inc.). Production of the vWF R1326H KI NSG mice was carried out by the Transgenic Mouse Model Core Facility.

**Clot retraction.** A total of 750 µl of Tyrode's buffer (127 mM NaCl, 5 mM KCl, 1.8 mM $NaH_2PO_4$, 1 mM $MgCl_2$, 25 mM $NaHCO_3$, 10 mM glucose, 0.1% $NaHCO_3$, pH 7.4) supplemented with inhibitor was mixed in a glass culture tube with 200 µl of human PRP and 5 µl red blood cells. Clotting was initiated by addition of 50 µl thrombin at 10 units/ml in saline and a sealed Pasteur pipette secured in the tube center. Digital photographs of the experiment were taken at 15-min intervals over 2 h. Images were analyzed with ImageJ software to determine the area occupied by the clot and plasma. Plots of the relative areas and linear regressions were performed with SigmaPlot (Systat Software, San Jose, CA).

**Cremaster arteriole laser injury model.** Human blood was collected in 0.129 M sodium citrate (10:1 vol/vol). Blood was obtained from healthy donors who provided written informed consent under a protocol that complied with all relevant ethical regulations and approved by the Children's Hospital of Philadelphia (CHOP) Internal Review Board in accord with the Helsinki Principles. PRP was separated after centrifugation (200×*g*, 10 min) at RT. The platelets were isolated from PRP, and prostaglandin E1 (Sigma-Aldrich) added to a final concentration of 1.0 µM. Platelets were pelleted by centrifugation (800×*g*, 10 min) at RT. The pellet was washed in calcium-free Tyrode's buffer (134 mM NaCl, 3 mM KCl, 0.3 mM $NaH_2PO_4$, 2 mM $MgCl_2$, 5 mM HEPES, 5 mM glucose, 0.1% $NaHCO_3$, and 1 mM EGTA, pH 6.5), and resuspended in CATCH buffer (phosphate-buffered saline (PBS) containing 1.5% bovine serum albumin, 1 mM adenosine, 2 mM theophylline, 0.38% sodium citrate, all from Sigma-Aldrich). Platelet counts were determined using a HemaVet counter (Drew Scientific). Animal experiments complied with all relevant ethical regulations, and were approved by the IACUC of the CHOP, and all investigators adhered to NIH guidelines for the care and use of laboratory animals. vWF^RH/RHNSG male mice were studied after being anesthetized using sodium pentobarbital (80 mg/kg) injected intraperitoneally and maintained under anesthesia with the same anesthetic delivered via a catheterized jugular vein at 5 mg/ml throughout the experiment. The thrombin-mediated microvascular injury model was performed as described[65]. Briefly, the cremaster muscle was surgically exteriorized and continuously superfused with PBS containing 0.9 mM $CaCl_2$ and 0.49 mM $MgCl_2$ maintained at 37 °C throughout the entire experiment. Human platelets ($4 \times 10^8$/mouse) were infused into the jugular vein with Alexa-488-labeled mouse anti-human CD41 F(ab′)$_2$ followed by Alexa Fluor-647 rat anti-mouse CD41 F(ab′)$_2$ (each at 3–4 µg/20 gm mouse) to detect

endogenous mouse platelets. Microvascular injury was induced with an SRS NL100 pulsed nitrogen dye laser (440 nm) focused on the vessel wall through the microscope objective. Each injury was followed for 3 min. Eptifibatide was used at 0.2 µg/g mouse body weight (BW; equivalent to the clinically effective dose[23]) and Hr10 was used at an equimolar concentration (2.4 µg/g BW). Drugs were infused 5 min prior to laser injury via the jugular vein. Pre and post drug measurements were made in the same animal. Wide-field images of the dynamic accumulation of fluorescently labeled platelets within the growing thrombi were recorded using a Hamamatsu ORCA Flash 4.0 V3 CMOS camera (Hamamatsu, Japan) coupled to an Excelitas X-Cite XLED light source. The microscope, cameras, and light sources were all controlled using Slidebook 6.0 software (Intelligent Imaging Innovations). Intensity of the fluorescent signal was used to measure incorporated platelets. Eight injuries were made in each of four mice per group. Studies in each animal were completed within 1 h after drug infusion to avoid the subsequent variable clearance of human platelets by host macrophages[66].

**Carotid artery thrombosis injury studies.** Redox-induced thrombosis was initiated in anesthetized vWF^RH/RHNSG post human platelet infusion ($8 \times 10^8$ human platelets) by Rose Bengal photochemical injury to the carotid artery as described[67]. Briefly, mice were first injected with 100 µl of PBS (Gibco) or PBS containing tirofiban, M-tirofiban or Hr10 via the jugular vein. Rose Bengal (50 mg/kg, Sigma-Aldrich) was then infused and a miniature Doppler flow probe (Model 0.5VB; Transonic Systems) was positioned around the artery. After 1 min of measuring baseline flow, a 540 nm laser was targeted to the carotid artery to induce an occlusive thrombus and blood flow was monitored for 40 min. Area under the curve of total blood flow was calculated. After 40 min of recording, the mice were euthanized.

**Tail bleeding studies.** Pentobarbital-anesthetized vWF^RH/RHNSG mice were infused retro-orbitally with $8 \times 10^8$ washed human platelets in a final volume of 200 µl (so that ~40% of circulating platelets were human). After 5 min, PBS containing 0.1% DMSO, or equipotent amounts of eptifibatide (0.2 µg/g BW), Hr10 (2.4 µg/g BW), tirofiban (0.025 µg/g BW), or M-tirofiban (0.32 µg/g BW) in the same buffer were administered intravenously. After another 5 min, the tip of the mouse tail (8 mm) was amputated with a sharp razor blade and the tail placed in a collection tube containing sterile water at 37 °C. To measure the total blood loss, the hemoglobin level in the water was quantified[68] with the following modifications: the hemolyzed whole blood/water mixture was centrifuged at 21,000×*g* in countertop Sorvall Legend Micro21 centrifuge (Thermo scientific) for 5 min. A total of 20 µl of clarified, stroma-free supernatant were diluted 10-fold in a 96-well microplate (Corning) and light absorbance was measured at 575 nm (Spectramax-190 plate reader, Molecular Devices). Blood loss during the 10-min window was measured based on standard curve previously obtained. Total bleeding time over the 10-min study was also recorded.

**Platelet clearance studies.** vWF^RH/RHNSG mice were anesthetized, injected with human platelets and drugs as in the photochemical carotid artery injury studies. Retro-orbital blood collection was performed preplatelet and drug infusions, and at time points. Collected blood was determined by flow cytometry using a combination of PE-conjugated rat anti-mouse CD41a and APC-conjugated mouse anti-human CD41a as above. Percent survival of human and mouse platelets was calculated. All statistics were done using nonparametric t-tests.

**Statistical calculations.** Dose–response experiments for whole blood aggregation and binding to K562 cells were conducted at least three times. Curve-fitting and statistical calculations were performed in SigmaPlot. The data points from each replicate were individually fit to a sigmoidal function to determine the minimum and maximum values for scaling. Data scaled to a maximum of 1 and a minimum of 0 were combined and again fit to a sigmoidal curve to determine the $IC_{50}$ value. The standard error for the $IC_{50}$ estimate was calculated using the reduced $\chi^2$ method. *p* values comparing $IC_{50}$s from different inhibitors were determined using the global fit function in SigmaPlot. The two data sets were fit with all parameters separate and again where the $IC_{50}$ value is shared between the data sets. Fisher's *F* statistic was calculated from the residual sum of squares and degrees of freedom for the unshared ($SS_{un}$, $DF_{un}$) and shared ($SS_{sh}$, $DF_{sh}$) with the equation $F = ((SS_{sh} - SS_{un})/(DF_{sh} - DF_{un}))/(SS_{un}/DF_{un})$ and the *p* value obtained from the *F* distribution. Linear regression fits to data from clot retraction experiments proceeded similarly. The Holm-Sidak test following one-way analysis of variance (alpha = 5.0%) was used to assess if the differences in human platelet accumulation in thrombi between Hr10 and eptifibatide-treated mice were significant. Each time point was analyzed individually, without assuming a consistent standard deviation. For the bleeding studies, the data passed the Shapiro-Wilk normality test and hence compared using the Student's *t*-test. Number of mice used for the bleeding studies is derived from the expectation of a large difference between treatments with eptifibatide or tirofiban, where only 5% of mice are expected to preserve normal hemostasis, and Hr10 or M-tirofiban, where we expect 80% would preserve normal hemostasis based on our clot retraction studies. With these assumptions four animals per group will produce a significance level of 0.05 with 90% statistical power[69].

**Reporting summary**. Further information on research design is available in the Nature Research Reporting Summary linked to this article.

## Data availability

All relevant data are included in the paper and/ or its supplementary information files. The source data of Figs. 2a–d; 4a–c, e, f; 5b, c; 6a, b; 7c, d; 8a–d; 9a–c; and Supplementary Figs. 3 and 5 are provided in a Source Data file. The atomic coordinates and structure factors for the reported crystal structure of αVβ3/Hr10 complex have been deposited in the Protein Data Bank (PDB) under the accession code 6NAJ. [http://www.rcsb.org/pdb/results/results.do?tabtoshow=Unreleased&qrid=AD6A3C5C], where they can be obtained free of charge.

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

## Acknowledgements

We thank Drs. Donald W. Landry, Shi-Xian Deng, and Alison Rinderspacher (Columbia University, NY, NY) for synthesis and chemical analysis of M-tirofiban, and Dr. Jennifer Cochran (Stanford University, CA) for providing αIIbβ3-K562. We also thank Drs. Mark Ginsberg (UC San Diego, CA), Thomas J Kunicki (Scripps Research Institute, CA), and Makoto Handa (Keio University, Tokyo, Japan) for providing mAbs LIBS-1, AP5, and PT-25-2, respectively. B-95ΔA cells and pCDF5-Har plasmid were kindly provided by Dr. Kensaku Sakamoto (RIKEN, Yokohama, Japan). This work was supported by NIH grants DK088327, DK48549, and HL141366 (to M.A.A.), DK101628 (to J.V.A.), and R01HL142122 and P01HL040387 (to M.P.) from the National Institutes of Diabetes, Digestive and Kidney diseases (NIDDK) and Heart, Lung and Blood (NHLBI) of the National Institutes of Health, a grant from the RICBAC Foundation (to M.A.A.) and grants from National Core Facility for Biopharmaceuticals, Ministry of Science and Technology, Taiwan (S.-W.L. and I.-S.Y.). A BioRxiv preprint version was first posted Dec 31, 2018.

## Author contributions

M.A.A. conceived and oversaw all experiments. B.D.A. and M.A.A. performed the aggregation and clot retraction assays. J.L.A. designed and performed the ligand and antibody binding studies. J.V.A. collected the diffraction data and refined the structures. J.V.A., J.X.P., and M.A.A. performed model building and structure analysis. S.-W.L. and I.-S.Y. generated the $vWF^{RH/+}$ NSG mice. V.H., H.S.A., and M.P. performed the mouse studies. All authors interpreted data. M.A.A. had the primary responsibility for writing the manuscript.

## Competing interests

Massachusetts General Hospital has submitted patent applications on the compounds described in this manuscript with M.A.A. named as inventor. The other authors declare no competing interests.
