## [Peer Review File · Nature Communications]

Reviewers' Comments:

Reviewer #1:

Remarks to the Author:

The platelet integrin α IIBb3 (glycoprotein IIB-IIIa) plays an essential role in the maintenance of normal hemostasis by regulating the formation of thrombi. Integrin α IIBb3 inhibitors have been widely assessed for therapeutic potential, resulting in several such drugs being approved for the treatment of coronary artery thrombosis. However, their clinical use has been shown to have serious side effects including bleeding. In this manuscript, the authors reported the structure-guided design of a ligand-mimetic peptide and a modified antagonist of drug Tirofiban, and showed that they inhibited human platelet aggregation but did not interfere with clot retraction. The discovery is very interesting and their data are strong and solid. However, a few questions need to be addressed:

1. M-Tirofiban was designed based on the Hr10/ α Vb3 and Tirofiban/ α IIBb3 structures, and the authors showed that M-Tirofiban bound with similar binding affinities to inactive and active α IIBb3, and blocked fibrinogen binding to preactivated α IIBb3. However, no experiment has been done on the binding of M-Tirofiban to the other b3 subfamily, α Vb3. In addition, the authors showed that Hr10 bound to both α IIBb3 and α Vb3, and the binding to α Vb3 was even better than that to α IIBb3 (Fig. 1b). However, the authors did not go further to determine the IC50 of Hr10 to α Vb3. It might be important to determine the binding affinities of both Hr10 and M-Tirofiban to α Vb3 since the binding might cause side effect when applying them as new anti-thrombosis drugs. In the discussion, the authors should also address this possibility.
2. It is interesting that both Hr10 and M-Tirofiban preserved clot retraction, in contrast to other α IIBb3 antagonists such as Eptifibatide and Tirofiban. The authors speculated that Eptifibatide or Abciximab inhibited clot retraction by inhibiting intracellular protein tyrosine dephosphorylation. However, since the binding of Hr10 or M-Tirofiban to α IIBb3 did not induce integrin conformational change, their binding should not affect intracellular signaling. The authors should also speculate whether the binding of Hr10 and M-Tirofiban to α Vb3 integrin, which is expressed in platelet, has any effect on preserving clot retraction.
3. The structure of α Vb3/hFN10 used in this manuscript was not clearly indicated. It is therefore very hard for the reviewer to evaluate the structure-guided design because of the missing information. In addition, all pdb files used should be clearly indicated both in the text and in the figure legends.

Reviewer #2:

Remarks to the Author:

The paper by Arnaout and collaborators is an interesting study showing the utility of structure based design in the realization of two new full antagonists of integrin α -IIB- β -3.

The authors are important players in the field and have contributed various crystal structures, including the one of the hFN10 complex with α -v- β 3.

I found the paper interesting, complete and well written.

I suggest two revisions that the authors should tackle to convey complete information on their design.

- 1) the authors should show the molecular structure of the modified tirofiban and the sequence of their Hr10 peptide in the main text. Otherwise it is difficult to appreciate the chemical details that define the potency of the two new molecules.
- 2) With regards to the design, the concepts of engaging the region close to the binding site with extended pi-pi interactions had been previously formalized also by the computational work of Paladino and coworkers in PLoS Comput Biol. 2017 Jan 23;13(1):e1005334; and in Chemistry. 2019 Apr 23;25(23):5959-5970. These papers and possibly others from other authors should be cited and included in the discussion on the development of the ligands.

Reviewer #3:

Remarks to the Author:

The manuscript entitled "Structure-guided design of novel orthosteric inhibitors of α IIB β 3 that prevent thrombosis but preserve hemostasis" described that structure-guided designed the ligand mimetic peptide Hr10 and a modified form of the partial agonist drug Tirofiban act as "pure" orthosteric antagonists of α IIB β 3. They claimed that both agents no longer induce the conformational changes in α IIB β 3 and also suppress these changes when induced by agonists. These two agents strongly inhibited platelets aggregation and thrombus formation without impairing hemostasis. They concluded that the pure orthosteric inhibitors of α IIB β 3 may thus offer safer alternatives for human therapy. The structure-guided approach may also find utility in designing similar drug candidates targeting other integrins and in providing vital tools for further probing structure-activity relationships in integrins. This is an interesting work and manuscript has been relatively well prepared. There are, however, some concerns that need authors to address:

Major concerns:

1) Figure 1b: what is the copy number of α IIB β 3 or α V β 3 on the cell surface? Are they activated or not? Why are only 40% cells positive for Hr10?

Figure 1d: Is there significant difference of AP5 binding after ADP treatment between No Inhibitor and Hr10? How Hr10 can inhibit AP5 binding and whether this can be reproduced using purified α IIB β 3 should be addressed. AP5 epitope is located in the N-terminus of the integrin beta3 subunit, which is far away from the ligand binding/MIDAS site. The purified α IIB β 3 (not on platelets, no intracellular signal events) may be able to provide more evidence to support the authors' claim "no longer induce conformational changes".

2) From the figure 3a-d, the authors claimed that Hr10 blocked platelet aggregation induced by the agonists collagen, ADP and TRAP as effectively as Eptifibatide (Fig. 3a-d). However, the mean IC50 of Hr10 for the ADP and TRAP induced platelet aggregation is two times more than that of Eptifibatide. Therefore the Eptifibatide seems to be a stronger than Hr10, the authors may re-think whether "as effectively as Eptifibatide" is an overstatement. The authors should test whether at low doses of agonists (ADP, collagen, and TRAP) this difference of platelet aggregation inhibition can be even more obvious.

Since the Hr10 has similar inhibition function as Eptifibatide in collagen induced aggregation, have the authors checked or excluded whether the Hr10 cross-react with α 2 β 1 (and/or GPVI), which may compensate its possible weaker inhibitory effect on α IIB β 3?

Figure 3 e and f: The authors may provide more explanation why Hr10 can inhibit dense granule release but enhance alpha granule release. ADP is usually less potent to induce platelet granule release. Can authors reproduce the results of alpha and dense granule releases after thrombin/TRAP and collagen treatments? In addition, if Hr10 can enhance the ADP induced P-selectin expression, it may be difficult to call it "pure antagonist".

3) Figure 4a, is there any specific reason for the authors not to use whole blood for the clot retraction? Can the weak inhibitory (it seems to not be the same as the control) effect on clot retraction by Hr10 in Figure 4a be explained by the similar mechanism for platelet aggregation (weaker inhibitory effect for platelet aggregation in figure 3 b and c as compared with Epti.)?

4) Figure 5b, why did the authors only show the bleeding volume without bleeding time (the most common test to show the bleeding disorder)?

In addition, since the tail bleeding assay may not well reflect the hemostasis, the authors should also employ other bleeding assays to test whether Hr10 indeed does not affect hemostasis, even

at higher doses.

5) The authors should also examine whether Hr10 induces platelet clearance in vivo in their mice?

Minor issues:

1) Page 2, Introduction (line 4), in addition to fibrinogen-mediated platelet aggregation, platelet aggregation and thrombus formation occur in mice deficient fibrinogen (i.e. fibrinogen-independent aggregation). The authors may introduce "fibrinogen or other proteins" for this sentence and cite the work (Yang H et al, J Thromb Haemost. 2006 Oct;4(10):2230-7).

2) Page3, the line 2 from the bottom, the "FB" should be defined in its first appearance in the manuscript.

3) The line 6, page5, the second "that" should be either "the" or "those"

Reviewer #1 (Remarks to the Author):

The platelet integrin $\alpha\text{IIb}\beta\text{3}$ (glycoprotein IIb-IIIa) plays an essential role in the maintenance of normal hemostasis by regulating the formation of thrombi. Integrin $\alpha\text{IIb}\beta\text{3}$ inhibitors have been widely assessed for therapeutic potential, resulting in several such drugs being approved for the treatment of coronary artery thrombosis. However, their clinical use has been shown to have serious side effects including bleeding. In this manuscript, the authors reported the structure-guided design of a ligand-mimetic peptide and a modified antagonist of drug Tirofiban, and showed that they inhibited human platelet aggregation but did not interfere with clot retraction. The discovery is very interesting and their data are strong and solid. However, a few questions need to be addressed:

1. M-Tirofiban was designed based on the Hr10/ $\alpha\text{V}\beta\text{3}$ and Tirofiban/ $\alpha\text{IIb}\beta\text{3}$ structures, and the authors showed that M-Tirofiban bound with similar binding affinities to inactive and active $\alpha\text{IIb}\beta\text{3}$, and blocked fibrinogen binding to preactivated $\alpha\text{IIb}\beta\text{3}$. However, no experiment has been done on the binding of M-Tirofiban to the other β3 subfamily, $\alpha\text{V}\beta\text{3}$. In addition, the authors showed that Hr10 bound to both $\alpha\text{IIb}\beta\text{3}$ and $\alpha\text{V}\beta\text{3}$, and the binding to $\alpha\text{V}\beta\text{3}$ was even better than that to $\alpha\text{IIb}\beta\text{3}$ (Fig. 1b). However, the authors did not go further to determine the IC_{50} of Hr10 to $\alpha\text{V}\beta\text{3}$. It might be important to determine the binding affinities of both Hr10 and M-Tirofiban to $\alpha\text{V}\beta\text{3}$ since the binding might cause side effect when applying them as new anti-thrombosis drugs. In the discussion, the authors should also address this possibility.

New data are now presented showing the IC_{50} of Hr10 binding to $\alpha\text{V}\beta\text{3}$ (Fig.1d) and of M-Tirofiban binding to $\alpha\text{V}\beta\text{3}$ (new Supplementary Figure 3). These data are now incorporated in the revised text (page 4, para 1 and page 6, para2) and the respective figure legends.

2. It is interesting that both Hr10 and M-Tirofiban preserved clot retraction, in contrast to other $\alpha\text{IIb}\beta\text{3}$ antagonists such as Eptifibatide and Tirofiban. The authors speculated that Eptifibatide or Abciximab inhibited clot retraction by inhibiting intracellular protein tyrosine dephosphorylation. However, since the binding of Hr10 or M-Tirofiban to $\alpha\text{IIb}\beta\text{3}$ did not induce integrin conformational change, their binding should not affect intracellular signaling. The authors should also speculate whether the binding of Hr10 and M-Tirofiban to $\alpha\text{V}\beta\text{3}$ integrin, which is expressed in platelet, has any effect on preserving clot retraction.

We agree that binding of Hr10 or M-Tirofiban to $\alpha\text{IIb}\beta\text{3}$ should not affect intracellular signaling. We have shown in new Suppl. Figure 3 that M-Tirofiban, which preserves clot retraction, does not bind $\alpha\text{V}\beta\text{3}$ (page 6, para2).

3. The structure of $\alpha\text{V}\beta\text{3}$ /hFN10 used in this manuscript was not clearly indicated. It is therefore very hard for the reviewer to evaluate the structure-guided design because of the missing information. In addition, all pdb files used should be clearly indicated both in the text and in the figure legends.

The crystal structure of the relevant ligand-binding region of $\alpha\text{V}\beta\text{3}$ /hFN10 is shown in green in Figure 1a. We have also indicated all the pdb files used throughout the text and figure legends.

Reviewer #2 (Remarks to the Author):

The paper by Arnaout and collaborators is an interesting study showing the utility of structure-based design in the realization of two new full antagonists of integrin α -IIb- β -3.

The authors are important players in the field and have contributed various crystal structures, including the one of the hFN10 complex with α -v- β 3.

I found the paper interesting, complete and well written.

I suggest two revisions that the authors should tackle to convey complete information on their design.

1) the authors should show the molecular structure of the modified tirofiban and the sequence of their Hr10 peptide in the main text. Otherwise it is difficult to appreciate the chemical details that define the potency of the two new molecules.

The molecular structure of the modified Tirofiban (new Fig.5a) and the sequence of the Hr10 peptide (new Fig. 1b) are now shown.

2) With regards to the design, the concepts of engaging the region close to the binding site with extended pi-pi interactions had been previously formalized also by the computational work of Paladino and coworkers in PLoS Comput Biol. 2017 Jan 23;13(1):e1005334; and in Chemistry. 2019 Apr 23;25(23):5959-5970. These papers and possibly others from other authors should be cited and included in the discussion on the development of the ligands.

Our 2014 *Nat Struct Mol Biol* paper (Van Agthoven et al, *Nat Struct Mol Biol* **21**, 383-388, doi:10.1038/nsmb.2797 (2014)) was the first to demonstrate by structure determination and mutational studies the importance of the key π - π stacking interaction between hFN10-W¹⁴⁹⁶ and Y¹²² in the ligand binding β A domain of α V β 3 in creating a pure orthosteric integrin inhibitor. In the revised version, we have quoted the confirmatory molecular dynamics study paper (page 3, para 2). The second paper focuses on binding the reverse ligand isoDGR, which is not referenced as the binding mode is qualitatively different from ligands used in the design of our inhibitors.

Reviewer #3 (Remarks to the Author):

The manuscript entitled "Structure-guided design of novel orthosteric inhibitors of α IIb β 3 that prevent thrombosis but preserve hemostasis" described that structure-guided designed the ligand mimetic peptide Hr10 and a modified form of the partial agonist drug Tirofiban act as "pure" orthosteric antagonists of α IIb β 3. They claimed that both agents no longer induce the conformational changes in α IIb β 3 and also suppress these changes when induced by agonists. These two agents strongly inhibited platelets aggregation and thrombus formation without impairing hemostasis. They concluded that the pure orthosteric inhibitors of α IIb β 3 may thus offer safer alternatives for human therapy. The structure-guided approach may also find utility in designing similar drug candidates targeting other integrins and in providing vital tools for further probing structure-activity relationships in integrins. This is an interesting work and manuscript has been relatively well prepared. There are, however, some concerns that need authors to address:

Major

concerns:

1) Figure 1b: what is the copy number of α IIb β 3 or α V β 3 on the cell surface? Are they activated or not? Why are only 40% cells positive for Hr10?

i) Recombinant α IIb β 3 and α V β 3 are expressed stably in equivalent numbers on the K562 cells. This is now shown in new Fig.1d, inset.

ii) The data in new Fig. 1c and Fig.1d show that Hr10 (or hFN10) can bind to the inactive integrin.

iii) The 40% cells positive for Hr10 in original Fig.1b was based on using an anti-His antibody to detect binding of the His-tagged Hr10 or hFN10. We suspected that the anti-His antibody sterically clashes with the α IIb helix, which is lacking in α V, giving the impression that Hr10 binds less to α IIb β 3 than to α V β 3. We therefore repeated this study using the directly fluoresceinated Hr10 and hFN10, which also allowed derivation of the respective IC₅₀s for these ligands (requested by Reviewer 1). These data, presented in new Fig.1c and Fig.1d, clearly show that Hr10 binds with higher affinity to inactive α IIb β 3 than to α V β 3. These data have been incorporated in the revised text (page 4, Para 1) and the respective Fig.1 legend.

Figure 1d: Is there significant difference of AP5 binding after ADP treatment between No Inhibitor and Hr10? How Hr10 can inhibit AP5 binding and whether this can be reproduced using purified α IIb β 3 should be addressed. AP5 epitope is located in the N-terminus of the integrin beta3 subunit, which is far away from the ligand binding/MIDAS site. The purified α IIb β 3 (not on platelets, no intracellular signal events) may be able to provide more evidence to support the authors' claim "no longer induce conformational changes".

i) Statistical significance of the level of AP5 binding after ADP between no inhibitor and Hr10 could not be determined as we lack replicates of the uninhibited condition due to problems with aggregation. The single determination is displayed in figure 1f in the revised manuscript.

ii) Our conclusion that Hr10 "no longer induces conformational changes" is based on using the identical approaches we used in our 2014 *Nat Struct Mol Biol* paper, namely, 1) the new crystal structure of α V β 3/Hr10 complex, which showed the integrin in the bent inactive state; 2) binding of the activation-sensitive mAb AP5 to the cellular integrin; and 3) binding of the extension-sensitive mAb LIBS-1 whose epitope is in the membrane-proximal leg domains of the cellular β 3 subunit. We avoid use of detergent-extracted full-length integrins in this setting, given the potential bias introduced by the heterogeneous integrin conformations of such preparations (*Coller BS, J Thromb Haemost. 2015;13 Suppl 1(S17-2)*).

2) From the figure 3a-d, the authors claimed that Hr10 blocked platelet aggregation induced by the agonists collagen, ADP and TRAP as effectively as Eptifibatide (Fig. 3a-d). However, the mean IC₅₀ of Hr10 for the ADP and TRAP induced platelet aggregation is two times more than that of Eptifibatide. Therefore the Eptifibatide seems to be a stronger than Hr10, the authors may re-think whether "as effectively as Eptifibatide" is an overstatement. The authors should test whether at low doses of agonists (ADP, collagen, and TRAP) this difference of platelet aggregation inhibition can be even more obvious.

We have revised the statement "as effectively as Eptifibatide" to now read: "Hr10 was as effective as

Eptifibatide in blocking collagen-induced platelet aggregation but was somewhat less effective in blocking ADP- or TRAP-induced aggregation.”(page 4, para 4).

Since the Hr10 has similar inhibition function as Eptifibatide in collagen induced aggregation, have the authors checked or excluded whether the Hr10 cross-react with $\alpha 2\beta 1$ (and/or GPVI), which may compensate its possible weaker inhibitory effect on $\alpha \text{IIb}\beta 3$?

We have not done so for two reasons: First, $\alpha 2\beta 1$ is not a member of the RGD-binding integrins (Hynes RO, Cell, 110, page 674, Fig. 1). So, $\alpha 2\beta 1$ (or GPVI) does not bind the RGD-based Hr10 or the FDA-approved drugs Eptifibatide and Tirofiban. Second, activation of $\alpha 2\beta 1$ on platelets requires preactivation of $\alpha \text{IIb}\beta 3$ (Van de Walle et al, Blood 109; 5095-602, 2007). It is expected that Hr10 will suppress $\alpha 2\beta 1$ indirectly by inactivating $\alpha \text{IIb}\beta 3$.

Figure 3 e and f: The authors may provide more explanation why Hr10 can inhibit dense granule release but enhance alpha granule release. ADP is usually less potent to induce platelet granule release. Can authors reproduce the results of alpha and dense granule releases after thrombin/TRAP and collagen treatments? In addition, if Hr10 can enhance the ADP induced P-selectin expression, it may be difficult to call it “pure antagonist”.

At saturating concentrations of both Hr10 and Eptifibatide, dense granule release but not α granule release was inhibited. This is consistent with the properties of current $\alpha \text{IIb}\beta 3$ inhibitors. Other data have shown that agonist-induced α -granule release from platelets is $\alpha \text{IIb}\beta 3$ -independent (new reference 31, Elaib et al. Blood 128:1129,2016)(page 5, para 1). These data make it both unnecessary to further test TRAP/collagen, and also do not impact on our statements that Hr10 is a pure $\alpha \text{IIb}\beta 3$ antagonist. The reviewer’s conclusion that Hr10 enhances α granule release is caused by our mislabeling the y-axis in Fig. 3f, which failed to indicate that the buffer sample did not contain ADP. This oversight has been corrected (Fig.3f).

3) Figure 4a, is there any specific reason for the authors not to use whole blood for the clot retraction? Can the weak inhibitory (it seems to not be the same as the control) effect on clot retraction by Hr10 in Figure 4a be explained by the similar mechanism for platelet aggregation (weaker inhibitory effect for platelet aggregation in figure 3 b and c as compared with Epti.)?

i) In the literature, clot retraction is standardized using PRP because red blood cells in whole blood impair retraction as a function of the level of hematocrit (see the already cited references by Tucker et al, 2012 and Tutwiler et al, 2017).

ii) Quantitative data in Fig. 4b show equivalent preservation of clot retraction in buffer vs. Hr10 ($p=0.125$), and Fig 1e showed that Hr10 has a higher binding affinity to $\alpha \text{IIb}\beta 3$ (30.3 ± 4.8 nM) than Eptifibatide (73.2 ± 7.0 nM). The concentration of all inhibitors is $1.5 \mu\text{M}$, which is more than 10 times the IC_{50} and demonstrated to saturate the receptors when assayed for aggregation (Fig 3a-c) and binding (Fig1c,e). So, a “weaker inhibitory effect” of Hr10 vs. Eptifibatide is not supported by these data.

4) Figure 5b, why did the authors only show the bleeding volume without bleeding time (the most common test to show the bleeding disorder)? In addition, since the tail bleeding assay may not well reflect the hemostasis, the authors should also employ other bleeding assays to test whether Hr10 indeed does not affect hemostasis, even at higher doses.

In addition to quantifying blood loss, we have now added the bleeding time test in the new Fig. 6c. and presented these data in the revised text (page 6, para 3) and figure legend. To our knowledge, there are no other established hemostatic mouse models to test.

5) The authors should also examine whether Hr10 induces platelet clearance in vivo in their mice? This has now been done and the data presented in new Supplementary Fig. 2, in the text (page 6, para 3), and the methods described in the Methods section (page 12 para 2).

Minor

issues:

1) Page 2, Introduction (line 4), in addition to fibrinogen-mediated platelet aggregation, platelet aggregation and thrombus formation occur in mice deficient fibrinogen (i.e. fibrinogen-independent aggregation). The authors may introduce "fibrinogen or other proteins" for this sentence and cite the work (Yang H et al, J Thromb Haemost. 2006 Oct;4(10):2230-7). We have now done so and added the reference (page 2, para 2).

2) Page3, the line 2 from the bottom, the "FB" should be defined in its first appearance in the manuscript.
Done.

3) The line 6, page5, the second "that" should be either "the" or "those"
Done.

Reviewers' Comments:

Reviewer #1:

Remarks to the Author:

The revised manuscript is now acceptable according to this reviewer's opinion.

Reviewer #2:

Remarks to the Author:

the authors have responded to the requests

Reviewer #3:

Remarks to the Author:

This is an interesting work and manuscript has been well prepared. The authors have adequately addressed my previous questions and no more question from this reviewer.

It will be, however, appreciated by the readers if the authors can provide more explanation how Hr10 can inhibit platelet aggregation but preserve thrombin-induced clot retraction.

In addition, since this manuscript will be likely published, their sentence "In this manuscript, we used this information to engineer peptides paving the way for potentially safer integrin-targeted medical therapies." (page 3, the last sentence of the Introduction) should be changed to "In this study, we used....." or "Here, we used".

REVIEWERS' COMMENTS:

Reviewer #1 (Remarks to the Author):

The revised manuscript is now acceptable according to this reviewer's opinion.

Thanks.

Reviewer #2 (Remarks to the Author):

the authors have responded to the requests

Thanks.

Reviewer #3 (Remarks to the Author):

This is an interesting work and manuscript has been well prepared. The authors have adequately addressed my previous questions and no more question from this reviewer.

Thanks.

It will be, however, appreciated by the readers if the authors can provide more explanation how Hr10 can inhibit platelet aggregation but preserve thrombin-induced clot retraction.

We have designated a long paragraph in the discussion to address this (page 7, para#2)

In addition, since this manuscript will be likely published, their sentence "In this manuscript, we used this information to engineer peptides paving the way for potentially safer integrin-targeted medical therapies." (page 3, the last sentence of the Introduction) should be changed to "In this study, we used....." or "Here, we used".

We changed "manuscript" to "study" (page 3, para#2).